# Leisure Activities and Change in Cognitive Stability: A Multivariate Approach

**DOI:** 10.3390/brainsci7030027

**Published:** 2017-03-01

**Authors:** Nathalie Mella, Emmanuelle Grob, Salomé Döll, Paolo Ghisletta, Anik de Ribaupierre

**Affiliations:** 1Cognitive aging lab, FPSE, University of Geneva, Geneva 1211, Switzerland; salome.doll@gmail.com; 2Group of Developmental and Differential Psychology, FPSE, University of Geneva, Geneva 1211, Switzerland; Anik.deRibaupierre@unige.ch; 3Methodology and Data Analysis Unit, FPSE, University of Geneva, Geneva 1211, Switzerland; Emmanuelle.Grob@unige.ch (E.G.); Paolo.Ghisletta@unige.ch (P.G.); 4Swiss Distance Learning University, Brig 3900, Switzerland; 5LIVES–Overcoming vulnerability: Life course perspectives, University of Geneva, Geneva 1211, Switzerland

**Keywords:** intraindividual variability, cognitive aging, leisure activities, longitudinal study

## Abstract

Aging is traditionally associated with cognitive decline, attested by slower reaction times and poorer performance in various cognitive tasks, but also by an increase in intraindividual variability (IIV) in cognitive performance. Results concerning how lifestyle activities protect from cognitive decline are mixed in the literature and all focused on how it affects mean performance. However, IIV has been proven to be an index more sensitive to age differences, and very little is known about the relationships between lifestyle activities and change in IIV in aging. This longitudinal study explores the association between frequency of physical, social, intellectual, artistic, or cultural activities and age-related change in various cognitive abilities, considering both mean performance and IIV. Ninety-six participants, aged 64–93 years, underwent a battery of cognitive tasks at four measurements over a seven-year period, and filled out a lifestyle activity questionnaire. Linear multilevel models were used to analyze the associations between change in cognitive performance and five types of activities. Results showed that the practice of leisure activities was more strongly associated with IIV than with mean performance, both when considering overall level and change in performance. Relationships with IIV were dependent of the cognitive tasks considered and overall results showed protective effects of cultural, physical and intellectual activities on IIV. These results underline the need for considering IIV in the study of age-related cognitive change.

## 1. Introduction

Normal aging is traditionally associated with the deterioration of a number of cognitive abilities, such as working memory, episodic memory, processing speed, or inhibition [1]. More recently, growing emphasis has been placed on the importance of intraindividual variability (IIV) in aging. IIV, referring to within task fluctuation of performance, is indeed considered a reliable indicator of both neural and cognitive integrity [2,3,4,5] and is highly sensitive to age-related change in cognitive functioning [6,7]. Higher levels of IIV have been reported in patients with neurological diseases, such as Parkinson disease, Alzheimer disease, or mild cognitive impairment (MCI) [6,8,9,10,11]. In normal aging, the level of IIV has been associated with subsequent cognitive decline [12,13], working memory capacities [14], later MCI [15] and even with proximity to death [16]. All these findings converge to suggest that IIV is a meaningful indicator of processing efficiency in aging, and may provide, as suggested by some authors, unique information above and beyond information provided by average performance [3,17]. Consequently, a deeper understanding of how lifestyle activities impact IIV in aging would bring additional useful information on the risk and protective factors of general cognitive decline. 

Research interested in the protective factors of cognitive aging has shown a relation between active lifestyle and cognitive levels of functioning, on the one hand, and the rate of cognitive aging, on the other hand (e.g., [18] for reviews, [19,20,21,22]). For example, comprehensive reviews reported that regular physical activity plays a significant role in dampening age-related decline and reduces risks of developing subsequent dementia [18,19,20,23]. Similarly, several reviews pointed to relationships between intellectually stimulating activities and enhanced cognitive levels [19,22], although results from a longitudinal study show less reliably preventive effects on cognitive decline (see [22] for review). Engagement in social activities has also been found to relate to higher levels of cognition and better maintenance of cognitive functioning in old age [19,24,25]. Some studies also reported benefits of artistic activities, such as dance, writing, music, theater, and painting, on cognition [21]. However, other studies have found no or little relationship between the practice of leisure activities and cognitive performance in older adults or age-related change in cognitive performance [26,27,28,29]. 

Various reasons can be advanced for such discrepancies, many of which are methodological: use of a longitudinal versus a cross-sectional paradigm, different ways of measuring activity (self-reported questionnaires or objective measures), different operational definitions of activity categories, different types of analyses of the relationship between activity engagement and cognitive performance. This last point has been addressed in past studies using dual-change model analyses [30,31]. Lövden et al. [31] reported that social participation influences subsequent changes in perceptual speed, but not the reverse. Similarly, Ghisletta et al. [30] showed beneficial effects of media and intellectually demanding activities on subsequent changes in perceptual speed and not the reverse, suggesting that engagement in these activities may lessen cognitive decline, and not the reverse. 

Several potential mechanisms of these protective effects have been suggested in the literature. Physical activities are thought to have direct effects on the brain, including cardiovascular, neurotrophic, or neuro-inflammatory processes (see [18] for review), that would enhance or maintain cognitive reserve [32,33,34]. Intellectually stimulating activities or social stimulations would also, on the basis of the use-it-or-lose-it assumption, have a positive impact on cognitive reserve in older age [32,33]. Beyond direct effects of exercising cognition, some authors also considered that lifestyle activities may stimulate cognition by a general mechanism of enrichment that may act through different non-cognitive mechanisms, such as attitudes, aspiration or beliefs [19]. 

Research interested in how lifestyle affects cognitive IIV and age-related change in IIV is very scarce and somewhat controversial too. Investigating the relationships between aerobic fitness and IIV in adolescents, Wu et al. [35] reported that higher-fit children are less variable in their response time and more accurate, while no significant relationship with mean reaction times were observed. Using a longitudinal design, Bielak et al. [36] investigated relationships between processing speed, semantic and lexical decision, and activity engagement in older adults. The authors observed differences according to leisure activity type: Intellectually demanding activities were related to lower IIV in all tasks, while physical activity was only associated to IIV in less challenging tasks. However, the authors also reported that associations were stronger with mean performance than with IIV. Concerning longitudinal change, very limited protective effects of activities were observed on change in mean performance over a six-year period; they are again very few to address change in IIV. More recently, in a cross-sectional functional magnetic resonance imaging (fMRI) study, Kimura et al. [37] investigated the role of moderate physical activity, assessed by an electronic device, on variability in a task-switching paradigm. They reported that brisk walking in older adults, in contrast with slower rhythm in walking, was associated to less IIV and to a prefrontal functioning closer to that of young adults. Results concerning the protective effects of leisure activities on cognitive IIV are thus inconclusive. Apart from the study of Bielak et al. [36], relationships are confined to one single task and one single activity. Thus, a more systematic approach is needed to clarify the potential benefits of considering this index in the study of cognitive aging and of the general functional relationships between lifestyle and cognition. 

The present study aimed at exploring the association between a comprehensive panel of leisure activities, including physical, intellectual, cultural, artistic and social activities, and age-related cognitive change over a six- to seven-year period, considering both change in mean performance and change in IIV in a large set of neurocognitive tasks. The main contribution of our study is to consider several cognitive tasks and several types of leisure activities in the same sample while exploring relationships with both mean performance and stability in cognitive functioning. Based on previous reports that stability in cognitive functioning may be a more sensitive measure to individual and group differences than averaged cognitive performance, we may assume that the practice of leisure activities would be more related to IIV in cognitive performance than to mean level of performance. 

## 2. Materials and Methods

### 2.1. Participants

We analyzed data from the Geneva Variability Study (GVS), a longitudinal study in which participants are tested every two to two and a half years, on an extensive battery of neurocognitive tasks, mostly reaction time (RT) tasks but not only, especially designed to explore IIV [14,38]. The present paper focuses on the first four waves of data collection, i.e., a six- to seven-year period (mean interval of 6.25 years, standard deviation (SD) = 0.4). Two-hundred and seventeen participants were recruited at the beginning of the study, out of which ninety-six individuals, aged 59–87 years at the first wave of testing (81% women), completed all four waves (see Table 1 for participants’ characteristics). 

Participants were community-dwellers living in the region of Geneva, Switzerland, recruited either from the Senior University of Geneva or through newspaper and association advertisements for elderly individuals. All participants were native French speakers or fluent in French and had normal or corrected-to-normal vision. The study was approved by the ethical committee of the Faculty of Psychology and Educational Sciences of the University of Geneva. All participants gave written informed consent and received a small amount of money as a compensation for their transportation costs.

### 2.2. Procedure 

At each of the four waves, the same tasks were administered in the same order, during two to three sessions lasting about 1.5 h each, at one-week interval. All tasks were individually administered on a DELL computer, using E-Prime (Psychology Software Tools, Pittsburgh, PA, USA) [41], in a small room of our laboratory, and included processing speed, inhibition, and working memory tasks, a health questionnaire and a leisure activity questionnaire, as well as control measures of visual acuity, fluid intelligence, and vocabulary (see [14,38] for a full description of the tasks). 

#### 2.2.1. Leisure Activity Questionnaire

The leisure activity questionnaire was designed to assess the frequency of diverse activities, on a scale going from 0 (never), 1 (several times a year), 2 (several times a month), 3 (once a week), 4 (several times a week), to 5 (every day). Five items were related to cardio-vascular physical activity (aerobic, dancing, cycling, swimming, and skiing/hiking); three items concerned cultural activities (cinema/theater/concerts, exhibition/museum, or conferences/university courses); three items concerned social activities (playing cards, groups of seniors/volunteers or visiting friends/family); three items related to artistic activities (playing music/singing, painting, acting), and two items related to intellectual activities (chess, crosswords/puzzles). In addition, for each type of activity, participants had the opportunity to complete additional activities. The questionnaire was administered in this form at the fourth wave of testing. An identical version of the questionnaire, but also asking about the practice of activities five years earlier, was introduced during the third wave. Information regarding the contemporaneous practice of activities was therefore available for the third and fourth waves of testing, and retrospectively for the first wave, but only for a subsample of 66 participants. Wilcoxon tests showed no significant differences in the practice of social, cultural, artistic and intellectual activities, indicating that engagement in these activities rated at the fourth wave of experimentation was similar to engagement at baseline. However, Wilcoxon tests showed a significant decrease in dynamic activities, mostly in the practice of hiking and skiing; other sportive activities were practiced at a similar frequency at baseline. Data were thus quite similar and we preferred relying on non-retrospective responses, collected over the full present sample and over the entire period. A description of the descriptive data is provided in the Appendix A (Table 5). Furthermore, it should be noted that some activities were probed by a larger number of items but a few were dropped because they proved to be ambiguous from the participants’ point of view (for instance reading; walking; computer-related activity). 

#### 2.2.2. Cognitive Tasks

Six RT tasks (nine conditions) of various complexity were administered: A simple reaction time (SRT) task, consisting in pressing a button when a signal stimulus appeared; two choice reaction time tasks, line comparison (LI) and cross-square (CS), consisting in pressing the key corresponding to the location of either the longest line (LI) or a cross changing into a square (CS); a digit symbol substitution test (DI); and two conditions of a letter comparison task: six letters (LC6) and nine letters (LC9); a computerized version of the Stroop Colour Word test, including three conditions: neutral trials (refer to symbols in any colour; STn), congruent trials (refer to colour words in the congruent colour, e.g., blue in blue; STc) and incongruent trials (refer to colour words in an incongruent colour, e.g., blue in yellow; STi). All these tasks contained a large number of trials, in order to reliably assess IIV (see [14,38] for a full description of the tasks).

Working memory. Verbal and spatial-verbal working memory abilities were assessed using two tasks: a reading span (RSpan) task and a Matrices task. In the RSpan task, series of two to seven sentences were presented and participants had to perform a semantic judgment. At the end of each series, they had to recall the last word of each sentence. In the Matrices task, participants had to recall words and their corresponding position in a 5 × 5 grid. Two to seven items were shown on the grid. The number of correct associations (Matrices) was used here as the score. In both tasks, the difficulty level was adjusted to the participant’s memory span (*n*) assessed beforehand, and accordingly, two versions of the task were presented to the participants: level n and *n* + 1 in the RSpan task and level *n* + 1 and *n* + 2 in the Matrices task. Levels of difficulty higher than participant’s span were used in order to allow studying IIV in their performance. The number of correct responses, rather than a classical dichotomous pass/fail response, was used. A full description of the tasks is provided in Mella et al. [14,38]. 

Additional control measures were assessed, which included subjective health, fluid (Raven Progressive Matrices; [39]) and crystallized (Mill Hill Vocabulary Test; [40]) abilities. 

### 2.3. Data Preparation

The frequency of activities was summed for each subscale to create a composite activity measure taking into account both the number and the frequency of activities. Descriptive data showed that the social, cultural and physical subscales differentiated well participants, so that the scores could be treated as continuous. However, intellectual and artistic activities displayed a typical zero-inflated distribution, where a large part of the sample reported no activity. This type of distribution is not modeled well if the variable is treated as continuous [42]. Thus, we treated these two variables as dichotomous indicators, discriminating, for each of them, between less than once a week and at least once a week. Furthermore, correlation analyses indicated that social, cultural and artistic activities correlated with one another, while no significant correlation was observed with and between physical and intellectual activities (see Table 2).

Only RTs for correct responses were considered. They were trimmed as follows: extremely fast responses (RTs below 150 milliseconds (ms) for SRT, LI and CS tasks; 200 ms for ST; and 500 ms for LC and DI tasks) and extremely slow responses (RTs above 1000 ms for SRT, 1500 ms for LI and CS tasks; 2000 ms for ST; 5000 ms for LC and 12,000 ms for the DI task) were removed. Cleaning of the data resulted in an average loss of 1.06% of the data (0.003%–3.79% depending on the task). This trimming procedure, if anything, should reduce the estimated IIV. Finally, for each task and at each experimental wave of testing, individual mean level of performance (iM) and intraindividual standard deviations (iSD) were computed. 

### 2.4. Statistical Analyses 

A series of multilevel linear models were used to assess change in iM and iSD of all cognitive tasks across the four waves. Each model defined an intercept that summarizes overall level of performance, and a slope that captures change across the four waves. The first series of models estimated both fixed and random effects of time, to study sample average change and individual variations around this average change, respectively. The second series of models further included demographic covariates: age, sex, and their interaction. The third and final series of models further included the effects of engagement in activities on both the intercept and slope of each iM and iSD index. The effects of the practice of physical, intellectual, social, cultural, and artistic activities were entered together, for each task/condition and each index (iM, iSD). Entering them last and simultaneously allowed controlling for the effects of age, sex, and their interaction, and correlations between activities were taken into account. Time was coded as months in study, with a zero value at wave one. Initial age was centered on the sample mean, and sex coded with a zero value for women. 

## 3. Results

Descriptive statistics for the results of the questionnaire are provided in the Appendix A
Table 5.

### 3.1. Mean Performance

• Change across the four waves

Significant positive fixed effects of time for SRT, LI, CS, LC6, STn, indicated that participants became slower over a six- to seven-year period in the simplest RT tasks. In addition, random effects for intercepts and for the rate of change were significant for all RT tasks (except for the rate of change in LC9), indicating significant individual differences in level of performance and in change. 

A significant fixed effect of time for RSpan indicated that participants became better in this task. The fixed effect of time was not significant in the Matrices. However, random effects were significant for intercept in both working memory tasks and for the rate of change in the Matrices task. 

• Demographic effects 

Significant positive initial age effects were observed on the intercept for LC6, LC9, STi, and STn, indicating that, overall, older participants were significantly slower than younger ones. In addition, positive effects of age on the rate of change for CS, DI, STi, STc, and STn showed that older participants underwent a more pronounced slowing in these tasks than younger ones. Effects of sex were significant on SRT and LC9 slopes, suggesting that men became slower than women in LC9 and faster than women in SRT. 

Significant negative age effects on the intercept for the Matrices indicated that, overall, older participants had poorer performance than younger ones in these tasks. A significant effect of sex on the intercept for Matrices indicated that men performed better overall than women in the visuo-verbal task. Concerning the rate of change, a significant effect of initial age was observed for RSpan, suggesting that older individuals declined more than younger ones in this task. In addition, significant effects of sex on the slope of RSpan showed that men declined more in verbal working memory ability than women. 

• Relationships between activities and cognitive performance 

All effects are displayed in Table 3. Given the complexity of this table, the same results are summarized in Table 4: only significant effects are displayed, and the signs of effects have been adapted according to their positive or negative effect on cognition. Practicing intellectual activities at least once a week was significantly associated to faster RTs in LC9 and to better performance in the RSpan task at baseline. The weekly practice of artistic activities was also significantly related to faster responses in the LI task and to better performance in the visuo-verbal working memory task (Matrices). 

• Relationships between activities and cognitive change 

The frequency of practice of physical activities was significantly negatively related to change in LI and CS, indicating that individuals practicing physical activities more than the average of the group showed less slowing in the choice RT tasks. Similarly, the frequency of cultural activity was negatively related to change in the DI task and positively related to change in the RSpan task, indicating that individuals practicing cultural activities more than the average of the group showed less slowing down in the DI task and less increase in performance in verbal working memory. 

### 3.2. Intraindividual Variability 

• Change across the four waves

Significant positive fixed effects of time for SRT, LI, CS and STn indicated that participants became significantly more variable in the simpler RT tasks. No significant fixed effects of time were observed on any other RT tasks. In addition, random effects were significant on intercepts for all tasks, and on the rate of change for LI, CS, STn, and STc. Neither fixed nor random effects of time were significant in any working memory task. However, random effects were significant for the intercept in all working memory tasks. 

• Demographic effects

Significant positive effects of age on the intercept for CS, LC6, LC9, and STi suggest that older individuals were more variable on these RT tasks than younger ones. In addition, significant negative effects of sex for LC6 indicated that men were less variable in this RT task than women. Results concerning the rate of change showed a significant negative effect of age for STi, indicating that older individuals became more variable in inhibition abilities. 

• Relationships between activities and cognitive performance

Practicing intellectual activities at least once a week was significantly associated to lower variability (iSD) responses in LI, STc, and visuo-verbal working memory at baseline. In addition, the weekly practice of artistic activities was significantly related to less variable RTs in LI, STc and STn. IIV in the RSpan task was also negatively related to the frequency of social activities and positively related to the frequency of cultural activities, indicating that greater IIV in verbal working memory was associated to less frequent social activities but to more frequent cultural activities. 

• Relationships between activities and cognitive change 

The frequency of physical activities was negatively related to change in IIV in LC6, suggesting that more frequent practice of physical activities was associated to less increase in IIV in this task. Similarly, the frequency of cultural activity was negatively related to change in IIV in LC6, LC9 and RSpan, indicating that more frequent cultural activities were associated to less increase in IIV in complex RT tasks and in verbal working memory. Greater social activities were also significantly related to less increase in IIV in verbal working memory. Practicing intellectual activities at least once a week was negatively related to change in IIV in STi, suggesting that it protects from an increase in IIV in this inhibition task. Lastly, practicing art at least once a week was positively related to change in IIV in LC9, that is, to an increase in IIV in this task. 

## 4. Discussion 

The present study aimed at exploring the relationships between lifestyle factors, notably physical, intellectual, social, cultural, and artistic activities and both cognitive abilities in older adults and age-related cognitive change over a six- to seven-year period, using a large battery of cognitive tasks and exploring cognitive performance both in terms of level of performance and in terms of IIV in performance. Consistently with our predictions, lifestyle activities were more strongly related to IIV in performance than to average level of performance, suggesting that they may have a more important role in the stability of cognitive functioning, than in global cognitive performance in aging. However, significant effects are less numerous than we had expected, and they vary depending on tasks and activities. Yet some trends emerge from these results: intellectual and artistic activities were more related to the initial level of performance, whereas physical and cultural activities were more related to change in performance. The frequency of social activities was almost never related to cognitive performance or to cognitive change. 

Practicing art at least once a week seems to be positively related to processing speed: It was associated to faster RTs in a choice RT task and to less variable RTs in three rather simple conditions (LI, STc, STn). In addition, weekly artistic practice was associated to better performance in visuo-verbal memory. In line with previous studies investigating the role of art activities, including expressive writing, music, theatre or visual art, in cognition (see [21] for a comprehensive review), our findings suggest that weekly art practice is related to enhanced cognitive functioning. Prior studies report that weekly practice of music was related to higher working memory performance and visuo-motor speed [43,44], which is consistent with enhanced/protective effect of art activities on visuo-verbal memory, although artistic activities in our experiment did not include only music. Other research interested in theater, based on intervention paradigms, has shown that four weeks of practice of acting significantly improved cognitive functioning and decreased risk factors for dementia in mentally healthy older adults [45,46,47]. Enhancing effects of art practice on cognition are thought to be mostly indirect. Artistic activities are hypothesized to positively activate the brain reward system, thereby promoting emotional well-being and motivation [48]. In addition, commitment in a creative process involves regular and disciplined practice over time and deep levels of concentration, which may have a training effect on focused attention abilities. This assumption may explain why art practice has more impact on IIV in processing speed than on average RTs: participants engaged in weekly art practice may more easily maintain attention in cognitive tasks. Unexpectedly however, practicing artistic activities weekly was associated to increased IIV in visuo--spatial working memory and in a task of complex RT. 

Weekly practice of intellectually stimulating activities was also related to better performances in inhibition and in verbal working memory, and to the overall level of IIV in two RT tasks and in visuo-verbal working memory. The important role of intellectual activities for cognition is not surprising, given their inherent nature to promote thinking and remembering processes. Quite a few longitudinal studies reported that higher engagement in intellectual activities was protective of cognitive decline, even when controlling for multiple covariates [49,19]). Yet, our results mostly displayed relationships between intellectual activities and the initial level of cognitive abilities; very few relationships with the rate of change were significant—with the exception of a positive relation with IIV in cognitive inhibition. Discrepancies with other studies may be explained by differences in measuring cognitive decline or abilities. For example, a number of studies were interested in a qualitative change in cognitive status, such as development of incident dementia or cognitive impairment [50,51,52,53,54], but did not assess change in a full spectrum of cognitive abilities, and very few were interested in healthy aging. Protective effects of intellectual activities on cognition may come from their brain-stimulating nature. According to some authors, such activities might allow for developing synaptic complexity and help in maintaining existing neuronal pathways, therefore enhancing brain plasticity and protecting from normal age-related neurodegenerative processes (e.g., [48]). This assumption was nevertheless dampened by other findings showing that, while brain structure integrity is positively related to frequency of practice of physical activities three years earlier, no such protective effects were observed for leisure activities [55]. Alternatively, and similarly to art activities, the weekly practice of intellectually stimulating activities may enhance general sustained attention abilities, which is again consistent with more relationships with IIV than with mean cognitive performance; interestingly, relationships with mean level of performance concerned cognitive inhibition and verbal working memory, two rather demanding tasks. 

Results showed positive relationships between physical activities and RT tasks: more frequent practice of physical activities was significantly related to less age-related slowing in the two choice RT tasks and to less increase in IIV in one out of three conditions assessing highly complex processing speed. Our findings are consistent with those of Bielak et al. [36], showing a protective effect of physical activities only in the choice RT task. They are also in line with longitudinal studies showing that better cardiorespiratory fitness or self-reported engagement in physical activities at baseline predicts subsequent cognitive change [56,57]. Unlike in prior studies [56], relationships with change in verbal working memory were not significant. However, our sample of older adults showed an intriguing positive change in verbal working memory, i.e., participants increased their performance over six–seven years, which renders difficult exploration of conditions associated with decline in this task. Physical activities were significantly related to change in processing speed, but not to baseline performance. Contrasted effects between rather simple versus complex processing speed tasks are interesting. Several studies indeed suggest that physical activities have direct effects on brain structure and function [55,58,59,60]. Gow et al.’s results [55] particularly suggest that physical activities have more direct effects than leisure activities on brain structure. Yet, a few magnetic resonance imaging studies suggest that IIV in processing speed relies, more than average performance, on white matter integrity [61,62,63]. A speculative hypothesis could be that relationships between physical activity and average processing speed, on the one hand, and IIV, on the other hand, might reflect a gradient of age-related decline in white matter integrity. Complex RT tasks might call for more distributed brain areas and a loss of white matter integrity in a few of these brain regions would result in more IIV in complex processing speed. Conversely, simple RT tasks call for basic processing speed processes, relying on less distributed brain areas. A more uniform loss of integrity in these brain areas would then result in a general slowing. This assumption is speculative and future functional and structural MRI studies are necessary to test it. 

Results concerning cultural activities were mixed, some showing positive relationships with cognitive change, some a negative association with cognition. Individuals engaging more frequently in cultural activities were also those showing less age-related slowing in one of the complex speed processing tasks (DI) and less increase in IIV in complex speed processing tasks and in verbal working memory. Engagement in cultural activities such as attending an exhibition or going to the museum seems thus to be more related to complex cognition, and, here again, more to IIV than to mean performance. Negative relationships with cognition were, however, also observed: more frequent cultural activities were associated to more variable verbal working memory performance, and to less increase in average performance in this task. As previously mentioned, results concerning the RSpan task are to be taken with caution, because of the unexpected improvement in this task over the considered period, which probably reflects effects of habituation or of developing strategies. In addition, these associations were very small (less than 0.001) and difficult to interpret. This is also the only task showing significant relationships with social engagement: greater social engagement was significantly associated to less variable performance and to less age-related increase in IIV in verbal working memory. Notwithstanding the above-mentioned difficulties to interpret findings relative to this task, one can hypothesize that it is in line with studies showing a positive effect of social network or social engagement on cognitive functioning [64]. However, they appear inconsistent with Bielak et al.’s reports of no relationship between social activity and IIV [36], the only study to our knowledge to have addressed this question. Discrepancy may result from the use of different tasks, all of which were based on RTs in Bielak et al.’s study, whereas in the present study, working memory has been scored in terms of accuracy. 

There are certain limitations to our study. The major limitation is that the activity measures were taken at the last wave of experimentation. As mentioned in a footnote, a subsample of the participants had to also rate retrospectively, at the time of the third wave, the intensity of their practice of the same activities five years earlier, that is, at baseline. Analyses showed no significant differences over the five years, with the exception of a decrease in hiking/skiing activities. Therefore, apart from dynamic activities, one may be confident that engagements were rather comparable at T4 and baseline. Yet, the question of the directionality of effects remains. The traditional and intuitive view is that lifestyle activities impact cognitive functioning, through previous experience, cognitive training, brain plasticity, or other non-physiological mechanisms. This view has been supported by prior longitudinal studies [30,31]. However, one might also consider an effect in the inverse direction, i.e., individuals feeling cognitively better might be more likely to engage in diverse activities. The design of our study does not allow for addressing this question and we can only speak in terms of associations between activities practice and cognitive functioning. Another limitation is linked to the number of items, somewhat small in some activity subscales; for instance, there were few items for the intellectual activities because a number of other items were ubiquitous and proved to be interpreted differently by different participants (e.g., an item about computer activities was obviously understood like basic activities such as using a smartphone rather than computer skills like programming, or an item about walking was also interpreted as a daily activity rather than as a sportive one); therefore, they had to be dropped. 

## 5. Conclusions

Altogether, our findings point to complex relationships between the practice of leisure activities, on the one hand, and cognitive abilities or age-related change in cognition, on the other hand. Considering the wide range of the cognitive tasks involved in our study, normal aging appears to present a fairly limited sensitivity to engagement in such leisure activities. However, physical and cultural activities presented stronger or more frequent associations with change in cognitive abilities, while intellectual and artistic activities showed more relationships with the initial level of performance. Importantly, given the objective of the present study, engagement in these activities was more often related to cognitive stability than to mean performance. This suggests a possible mechanism of maintenance/enhancement of sustained attention abilities associated to engagement in leisure activities. Our results also point to the fact that IIV in cognitive functioning is a useful index of age-related cognitive change, not redundant with the mean performance, which may overlook other facets of cognitive functioning. 

## Figures and Tables

**Table 1 brainsci-07-00027-t001:** Participants’ characteristics (*n* = 96).

	Education ^1^	Age	Fluid intelligence ^2^	Vocabulary ^3^
	Mean (SD)			
Wave 1	14.78 (3.44)	68.15 (6.05)	38.38 (8.29)	28.60 (4.14)
Wave 4	14.78 (3.44)	74.46 (6.18)	37.98 (9.46)	28.35 (4.50)

^1^ Years of education; ^2^ Raven Progressive Matrices [39]; ^3^ Mill Hill Vocabulary Test [40]. SD = standard deviation.

**Table 2 brainsci-07-00027-t002:** Correlations between practices of the different activities.

	Physical	Intellectual	Social	Cultural
Intellectual	−0.03			
Social	−0.02	0.00		
Cultural	0.14	−0.04	0.25 *	
Artistic	0.09	−0.15	0.26 *	0.38 **

* *p* < 0.05 (bilateral); ** *p* < 0.01 (bilateral).

**Table 3 brainsci-07-00027-t003:** Relationships between activity type and cognitive performance at baseline and subsequent change (mean performance and intraindividual variability (IIV)).

	Physical		Intellectual		Social		Cultural		Artistic	
**Mean**	**Inter.**	**Slope**	**Inter.**	**Slope**	**Inter.**	**Slope**	**Inter.**	**Slope**	**Inter.**	**Slope**
SRT ^1^	−1.17	−0.02	6.10	−0.09	−4.26	0.01	3.47	−0.07	−17.46	0.14
LI ^2^	0.44	−0.04 *	−7.49	−0.05	−1.69	0.03	4.50	−0.05	−42.18 **	0.21
CS ^3^	−0.81	−0.05 **	−1.00	−0.10	−0.58	0.01	3.16	−0.01	−27.13	−0.08
DI ^4^	−0.28	−0.05	−62.31	−0.73	−13.31	0.07	0.16	−0.27 *	−69.50	0.74
LC6 ^5^	15.11	−0.29	37.67	−0.30	5.39	0.01	−9.11	−0.41	−73.98	0.13
LC9 ^6^	35.26	−0.28	−143.06	2.11	36.48	−0.07	−32.25	−0.64	−218.71	2.54
STi ^7^	3.65	0.001	−57.60*	0.05	5.24	0.03	5.67	−0.07	−56.86	0.09
STc ^8^	2.33	0.01	−30.97	0.22	1.72	0.03	4.26	−0.08	−32.06	0.11
STn ^9^	1.37	0.01	−18.02	0.05	2.96	0.02	1.59	−0.06	−35.94	0.15
RSpan ^10^	0.02	−0.000	0.44 **	0.000	−0.02	0.000	0.02	−0.001 *	0.09	0.000
Matrices ^11^	0.01	0.000	0.14	0.003	0.000	−0.000	−0.01	−0.000	0.26 *	−0.004
**IIV**	**Inter.**	**Slope**	**Inter. **	**Slope**	**Inter.**	**Slope**	**Inter. **	**Slope**	**Inter. **	**Slope**
SRT	−0.58	−0.002	−2.77	−0.05	−0.79	0.01	0.87	−0.003	−6.36	0.01
LI	0.59	0.02	−12.87 **	−0.05	−0.39	0.02	0.66	−0.03	−12.70 *	0.06
CS	−0.18	−0.02	−7.00	−0.14	0.04	−0.002	0.55	−0.01	−8.10	−0.01
DI	−0.70	0.01	−26.58	−0.38	−1.27	−0.01	1.88	−0.03	−13.19	−0.17
LC6	7.94	−0.19*	−13.15	−0.55	−1.99	0.06	10.68	−0.30 *	14.81	0.01
LC9	14.61	−0.21	−89.80	0.80	4.98	0.02	6.53	−0.49 **	−112.23	2.39 **
STi	1.19	0.01	−8.05	−0.21 *	0.74	0.002	0.48	0.01	−18.31	−0.03
STc	0.87	0.01	−14.15 *	−0.02	0.56	−0.02	−0.27	0.01	−17.86 **	0.09
STn	1.16	0.003	−8.02	−0.06	0.78	−0.01	−0.08	0.01	−19.44 *	−0.003
RSpan	−0.002	−0.000	−0.01	0.000	−0.01 *	0.000 *	0.02 *	−0.000 *	−0.06	−0.000
Matrices	0.002	−0.000	0.10 *	−0.001	−0.004	−0.000	0.01	−0.000	−0.01	0.001

Note. * *p* < 0.05 (bilateral); ** *p* < 0.01 (bilateral). ^1^ Simple Reaction Time; ^2^ Line Comparison; ^3^ Cross-Square; ^4^ Digit Symbol; ^5^ Letter Comparison (six letters); ^6^ Letter Comparison (nine letters); ^7^ Stroop Incongruent; ^8^ Stroop Congruent; ^9^ Stroop Neutral; ^10^ Reading Span; ^11^ Matrices.

**Table 4 brainsci-07-00027-t004:** Synthesis of significant relationships between activity type and cognitive performance at baseline and subsequent change (mean performance and IIV).

	Physical		Intellectual		Social		Cultural		Artistic	
**Mean**	**Inter.**	**Slope**	**Inter.**	**Slope**	**Inter.**	**Slope**	**Inter.**	**Slope**	**Inter.**	**Slope**
SRT ^1^										
LI ^2^										
CS ^3^										
DI ^4^										
LC6 ^5^										
LC9 ^6^										
STi ^7^										
STc ^8^										
STn ^9^										
RSpan ^10^										
Matrices ^11^										
**IIV**	**Inter.**	**Slope**	**Inter. **	**Slope**	**Inter.**	**Slope**	**Inter. **	**Slope**	**Inter. **	**Slope**
SRT										
LI										
CS										
DI										
LC6										
LC9										
STi										
STc										
STn										
RSpan										
Matrices										

Note. Dark color is used to illustrate significant beneficial relationships of the types of activities with cognitive measures (faster RTs, better performance or lower IIV), *p* < 0.05. Light color is used to illustrate detrimental relationships on cognitive measures (slower RTs, poorer performance or increased IIV). ^1^ Simple Reaction Time; ^2^ Line Comparison; ^3^ Cross-Square; ^4^ Digit Symbol; ^5^ Letter Comparison (six letters); ^6^ Letter Comparison (nine letters); ^7^ Stroop Incongruent; ^8^ Stroop Congruent; ^9^ Stroop Neutral; ^10^ Reading Span; ^11^ Matrices.

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
