# Peer review of "Leisure Activities and Change in Cognitive Stability: A Multivariate Approach"

_brainsci, 2017, doi:10.3390/brainsci7030027_

Round 1
Reviewer 1 Report
The authors have conducted an interesting longitudinal study of the relationship between intra-individual variability on a variety of cognitive tasks and level of participation in a variety of leisure time of activities. The paper is well written in general (careful proofreading would be beneficial), the analyses are appropriate, and the results receive thorough interpretation in the discussion.
Concerns:
1. The authors are correct that more significant relationships between growth parameters and leisure activities were found for IIV (15) than for mean performance (7). However, 130 associations between intercept or slope and leisure activities were investigated for both mean performance and IIV. Thus, at most 10% of the associations achieved significance, and if we consider the more conservative significance level of p<.01, the numbers are much smaller: 2 for both mean and IIV. As a result, it seems an overstatement for the authors to conclude that “the course of normal aging is plastic and sensitive to the practice of leisure and physical activities.” These results suggest that sensitivity is, in fact, fairly limited.
Author Response
We thank the referee for his/her comment on our manuscript. We did our best to clarify the point that was highlighted.
This comment makes echoes to the second point of the other reviewer. New analyses introducing all activities in the same model were run, which had two effects: (1) It allowed controlling for the potential correlations between activities and (2) it considerably reduced the number of tests (from 90 to 13; or from 180 to 26 if we consider both IIV and iM) and therefore type-1 errors. Results showed no noticeable differences with previous ones. We chose to report results significant to p<.05 and p<.01, not corrected for multiple comparisons, because of the large number of tasks investigated in our study. This is an inherent and recurrent problem of studies using a large set of cognitive tasks. Adopting such a multivariate design is not the standard in the literature. On the one hand, our design allow studying cognitive performance / change in a more complete and complex way in the same individuals; on the other hand, having a large number of dependent variables increases the risk of type-1 errors. We considerably reduced of the number of tests, and most of effects were still significant; we hope that this will convince the reviewer to consider our results as reliable. We agree that the conclusive sentence concerning the plasticity of the course of normal aging was a little overstated and reformulated the entire conclusion, and the tone of the discussion.
Reviewer 2 Report
the leisure activities were measured at 4th wave, while the cognitive function was measured across waves. How can the leisure activities in a later time point predicted the cognitive function from baseline to the later time point?
for the multiple cognitive domains that are related to each other, the analytical results need to be adjusted for multiple comparison.
there have been accumulative literature on the relationship between leisure activities and cognitive function in aging population. It is unclear how the current study contributes to the existing literature. Studying a different format of cognitive aging using intra-individual variability did not provide sufficient rationale.
Author Response
We thank the referee for his/her comments on our manuscript. We did our best to clarify the points that were highlighted. Please find attached detailled replies to your comments.
Best regards
